# The First Detection of *Kudoa hexapunctata* in Farmed Pacific Bluefin Tuna in South Korea, *Thunnus orientalis* (Temminck and Schlegel, 1844)

**DOI:** 10.3390/ani10091705

**Published:** 2020-09-21

**Authors:** Gyoungsik Kang, Kwang-Min Choi, Dong-Hee Cho, Min-Soo Joo, Min-Jin Heo, Won-Sik Woo, Chan-Il Park

**Affiliations:** Department of Marine Biology and Aquaculture, College of Marine Science, Gyeongsang National University, Tongyeong 53064, Gyeongsangnam-do, Korea; gyoungsikkang@gmail.com (G.K.); tricolo1229@naver.com (K.-M.C.); nhjhyh@hanmail.net (D.-H.C.); fkffkxodn@hanmail.net (M.-S.J.); benny0911@naver.com (M.-J.H.); dnstory@hanmail.net (W.-S.W.)

**Keywords:** first detection, Pacific bluefin tuna, *Thunnus orientalis*, *Kudoa hexapunctata*, foodborne disease

## Abstract

**Simple Summary:**

In this study, we detected *Kudoa hexapunctata* in Pacific bluefin tuna (*Thunnus orientalis*) individuals that did not show any gross pathology lesions. Giemsa staining was used to identify clearer lesions than haematoxylin and eosin (H&E) staining that is used in general histological analysis. *K. hexapunctata* was separated through molecular biological methods, other than haematolocial and histological analysis. Individuals infected with *K. hexapunctata* showed relatively low haemoglobin (Hb) and haematocrit (Ht) values, and histological analysis revealed clear pseudocysts in the abdominal and dorsal muscles.

**Abstract:**

The consumption of fish and shellfish worldwide is steadily increasing, and tuna is a particularly valuable fish species. However, infection caused by *Kudoa* spp. is causing problems in many fish including the Pacific bluefin tuna (*Thunnus orientalis*), and there is much controversy about the association of these infections with foodborne disease. In this study, using haematological and histological analyses of the blood and internal organs (liver, spleen, kidney, heart, stomach, intestine, gill, and muscle) of Pacific bluefin tuna cultured in South Korea, infection with Myxosporea was first identified, and molecular biological analysis was conducted. In this study, *Kudoa hexapunctata* was finally identified. The Pacific bluefin tunas analysed in this study did not show any gross pathology lesions, such as visible cysts and/or myoliquefaction, of infection with this species. The histological analytical results can provide guidelines for the identification of *K. hexapunctata*. In the case of *K. hexapunctata*-induced infection, unlike other countries, such as Japan, there have been no reports in South Korea, and this study is the first to detect *K. hexapunctata* infection in Pacific bluefin tuna cultured in South Korea. The correlation between *K. hexapunctata* and food poisoning is not yet clear, however, it is thought that continuous observation of its infection is necessary.

## 1. Introduction

Because fish and shellfish products have exhibited nutritive benefits and because of the increasing demand for healthy food, the consumption of fish and shellfish products is increasing annually [1,2]. Among fish, tuna has a very high commercial value in many countries [3]. However, parasitic diseases in fish can cause foodborne diseases in humans, according to the literature, which indicates that *Kudoa septempunctata* is a potential threat to human health [4,5,6]. Additionally of interest, are *K. septempunctata* in olive flounder (*Paralichthys olivaceus*), *K. hexapunctata* in tuna, and their possible implication in causing foodborne diseases [4,5,6]. Furthermore, Japanese patients exhibiting clinical diarrhoea had eaten tuna in which *Kudoa hexapunctata* was detected [7]. According to Suzuki et al. [7], *K. hexapunctata* is likely to be one of the causes of foodborne disease; however, there is still no clear evidence. In Japan, cases of patients with “unidentified foodborne disease” have increased recently, and it is emerging as a serious problem [8]. In olive flounder, Ohnishi et al. [9] said that *K. septempunctata* can invade human intestinal epithelial cells e.g., Caco-2 cells and exhibit sporoplasm invasion, causing severe damage. From this point of view, *K. hexapunctata* can be seen as one of the most important agents of parasitic disease in tuna, due to its possibility to cause a zoonotic parasitosis.

*Kudoa* spp. has several types of spores which can be divided into various types based on the shape of their spore valves [10]. The diagnostic methods involve mainly PCR (polymerase chain reaction) and real-time PCR methods using 18S rDNA or 28S rDNA [10]. However, currently, studies are focused on how *Kudoa* spp. affect the human body, and techniques for their detection, as it is difficult to determine the shape and distribution of *Kudoa* spp. [11].

Here, we demonstrate that the detection of *K. hexapunctata* in the cultured Pacific bluefin tuna of South Korea, is accompanied by histological and molecular biological results. In particular, histological analysis can be applied to samples without gross pathology lesions and can be used to monitor *K. hexapunctata* infection in individuals without visible cysts and myoliquefaction, which are typical gross pathology lesions of *Kudoa* spp. infection [12,13,14]. Therefore, this study aims to provide basic data so that various experimental methods can be applied to detect *K. hexapunctata* infection by combining the two above-mentioned assays with a molecular biological method and is the first report on *K. hexapunctata* infection in South Korean cultured Pacific bluefin tuna (*Thunnus orientalis*). Furthermore, through this study, we intend to provide a resource to suggest *K. hexapunctata* infection in South Korean farmed tuna and to analyse the correlation between the source of infection and food poisoning.

## 2. Materials and Methods

### 2.1. Sample Preparation

Five farmed Pacific bluefin tunas (*Thunnus orientalis*) were randomly sampled in December 2019 on Yokji Island for a health condition assessment. Samples of blood from the heart were taken from each sample, and the internal organs (liver, spleen, kidney, heart, stomach, intestine, gill, and muscle) were harvested. Additionally, for histological analysis, each organ was placed in 10% neutral-buffered formalin and for molecular biological analysis, each organ was stored at −80 °C until analysis.

Ethical approval: All experimental protocols followed the guidelines of the Institutional Animal Care and Use Committee of the Gyeongsang National University (approval number: 2020-0002).

### 2.2. Condition Factor

Following Barnham and Baxter [15] and Biswas et al. [16], the condition factor (CF) was calculated as follows: CF = $10^{2}\times \frac{W}{L^{3}}$, where *W* = weight (g) and *L* = length (cm).

After calculation, CF values were compared Table 1 between each tuna, and the average condition of the tunas was calculated.

### 2.3. Morphological Analysis

For the morphological observation, abdominal and dorsal muscles (1 g) were sampled and homogenized using a 40 μm nylon cell strainer (Falcon, NY, USA). Staining was performed using Giemsa stain solution (Sigma-Aldrich, St. Louis, MI, USA) after the homogenates were suspended in 3 mL of PBS. For microscopic analysis, homogenates were smeared on a glass slide after being washed by distilled water.

### 2.4. Haematological Analysis

Each blood sample was collected in a heparin tube (BD, USA) for anticoagulation until arrival at the laboratory. The samples were centrifuged at 7000 RPM for 7 min, and the supernatants were used for haematological analysis by a DRI-CHEM 4000i instrument (Fujifilm, Tokyo, Japan). The following haematological parameters were analysed: alkaline phosphatase (ALP), blood urea nitrogen (BUN), calcium (Ca), glucose (GLU), glutamic oxaloacetic transaminase (GOT), glutamic pyruvic transaminase (GPT), lactate dehydrogenase (LDH), total cholesterol (TCHO), and total protein (TP). In the case of Ca and LDH, samples were diluted 2- and 10-fold in PBS for analysis. Haemoglobin (Hb) was measured using a haemoglobin colorimetric assay kit (Cayman Chemical, Ann Arbor, MI, USA) following the manufacturer’s protocol. Haematocrit (Ht) was calculated using heparinized capillary tubes (Paul Marienfeld Gmbh & Co. Kg, Lauda-Königshofen, Germany), and samples were placed in a haematocrit centrifuge machine (Scinco, Seoul, South Korea) at 12,000 RPM for, 5 min.

### 2.5. Histological Analysis

Dissection of internal organs (liver, spleen, kidney, heart, stomach, intestine, gill, and muscle) was followed by their fixation in 10% neutral-buffered formalin for 2 days. Then, small samples were taken from each organ, which were re-fixated in the same solution over 1 day before being gradually dehydrated with ethanolic solutions increasing from 70% to 100% ethanol. Samples were cleared further using xylene, after which, the samples were embedded in paraffin wax and, sectioned into slices with a 4 μm thickness. Finally, the sections were stained with haematoxylin-eosin (H&E) and Giemsa following routine protocols.

### 2.6. DNA Extraction, Polymerase Chain Reaction (PCR), and Sequence Analysis

Genomic DNA of tuna samples were extracted using an AccuPrep^®^ genomic DNA extraction kit (Bioneer, Daejeon, South Korea) following the manufacturer’s guidelines. Homogenized muscle sample (10 mg) was used for polymerase chain reaction (PCR) to detect *Kudoa* spp. by histological data. All genomic DNA samples were stored at −80 °C until analysis. PCR was performed for amplifying 16S rRNA and 28S rRNA gene. Briefly, 10 μL of Exprime Taq Premix (GeNet Bio, Nonsan, South Korea), 7 μL of distilled water, 1 μL of genomic DNA, and 1 μL of each forward and reverse primer were mixed. Sequences of primers and PCR conditions are displayed in Table 2.

For sequencing, amplicons were extracted using a QIAquick^®^ gel extraction kit (Qiagen, Hilden, Germany) following the manufacturer’s protocol. The purified PCR products were cloned into a pGEM^®^ T-easy vector (Promega, Madison, Wisconsin, USA), and transformed into Escherichia coli JM109 according to a general protocol. After full propagation, plasmid DNA was extracted using a Hybrid-Q™ plasmid rapidprep kit (GeneAll^®^, Seoul, South Korea) and sequenced using a universal M13 primer set. Nucleotide sequence matching was performed using the basic local alignment search tool (BLAST) algorithm of the National Centre for Biotechnology Information (https://blast.ncbi.nlm.nih.gov/blast).

## 3. Results

### 3.1. Condition Factor and Haematological Data

Five condition factor (CF) values of Pacific bluefin tunas and haematological results are presented in Table 3. *K. hexapunctata* was detected in one tuna (number 5), which had only low haemoglobin (Hb) and haematocrit (Ht) levels of 10.9 g/dL and 34%, respectively, compared to those in the other tuna samples with an average of 15.02 g/dL (Hb) and 44.5% (Ht). However, the results have no statistical significance because of the number of samples.

### 3.2. Morphological and Histological Analysis

The identified *K. hexapunctata* displayed six spores clearly (Figure 1). Histologically, the fifth sample only had pseudocysts in the abdominal and dorsal muscle (Figure 2 and Figure 3). By contrast, in the muscle of samples one to four, no pseudocysts indicating *K. hexapunctata* infection were observed in either the abdominal or dorsal sides. Similarly, liver, spleen, kidney, heart, stomach, intestine, and gill samples, including those of the fifth sample, did not display any evidence of *K. hexapunctata*-associated lesions.

### 3.3. Polymerase Chain Reaction (PCR) and Sequence Analysis

Ribosomal DNA (rDNA) nucleotide sequencing was performed using two sets of primers to detect *Kudoa* spp. [6,17,19]. First, the pseudocysts were considered as kudoid parasites, and universal primers were used for the first isolation [17,19]; the primer set reported by Arai et al. [6] was used for secondary diagnosis, which distinguished species of *Kudoa* spp. from pseudocysts observed in histological data (Figure 2 and Figure 3). As a result, electrophoresis showed a target band size (197 bp) and sequences that ultimately were associated with *K. hexapunctata* (Figure 4 and Table 4).

## 4. Discussion

The Pacific bluefin tuna (*Thunnus orientalis*) used in this study did not shown any common gross pathology lesions, such as white or yellow cysts or myoliquefaction in the abdominal and/or dorsal muscle, that have been seen in olive flounder (*Paralichthys olivaceus*) infected by *Kudoa septempunctata* [20]. Nevertheless, we finally identified *Kudoa hexapunctata* using histological and molecular biological analyses. Although, it is generally known that *K. hexapunctata* does not cause myoliquefaction, further detailed studies are needed to investigate the correlation between the *K. hexapunctata* and myoliquefaction [6].

*K. hexapunctata* and *Kudoa neothunni* share many single nucleotide polymorphisms (SNPs) in the 28S rRNA gene [6]. In addition, Arai et al. [6] also detected *K. neothunni* in Pacific bluefin tuna, which is generally considered to be an infection of yellowfin tuna. Because of its host specificity, a molecular biological method was used to distinguish between *K. hexapunctata* and *K. neothunni* in this study. It is true that histological examination provides a guideline for definitive diagnosis; nevertheless, it can be seen that molecular biology methods have been included for accurate analysis.

In the present study, we investigated *K. hexapunctata*-infected tuna’s biomass and haematological data. It could be confirmed that the condition factor (CF), haemoglobin (Hb), and haematocrit (Ht) values of the fifth sample were relatively low, but the result was not statistically significant because the number of samples was small. Histologically, there were no specific pathological lesions such as necrosis or edema in other organs; nevertheless, further study on this part is needed. Moreover, on H&E staining, it was easy to identify myxosporea; however, the characteristic structure of *K. hexapunctata* was difficult to observe, and the spores of *K. hexapunctata* were much easier to observe through Giemsa staining.

There are no reports of South Korean tuna consumers complaining of food poisoning symptoms such as vomiting or diarrhoea; however, the causal relationship between *K. hexapunctata* and foodborne disease needs to be investigated in more depth by steadily checking the *K. hexapunctata* infection of tuna distributed in South Korea. Through this study, we confirmed that *K. hexapunctata* exists in cultured Pacific bluefin tuna in South Korea, and it is thought that there is a need to provide safer food resources to consumers through continuous surveillance of *K. hexapunctata*.

## 5. Conclusions

In conclusion, this is the first report of the detection of *Kudoa hexapunctata* from cultured Pacific bluefin tuna (*Thunnus orientalis*) in South Korea. The subjects used in this analysis had no visually identifiable gross pathology lesions such as visible cysts or myoliquefaction. The association between *Kudoa* spp. induced infection and food poisoning has been reported steadily in olive flounder (*Paralichthys olivaceus*) and Pacific bluefin tuna, but there is no clear evidence yet. Therefore, further studies need to test the potential pathogenicity in laboratory mammals for the purpose of speculating about possible foodborne disease.

## Figures and Tables

**Figure 1 animals-10-01705-f001:**
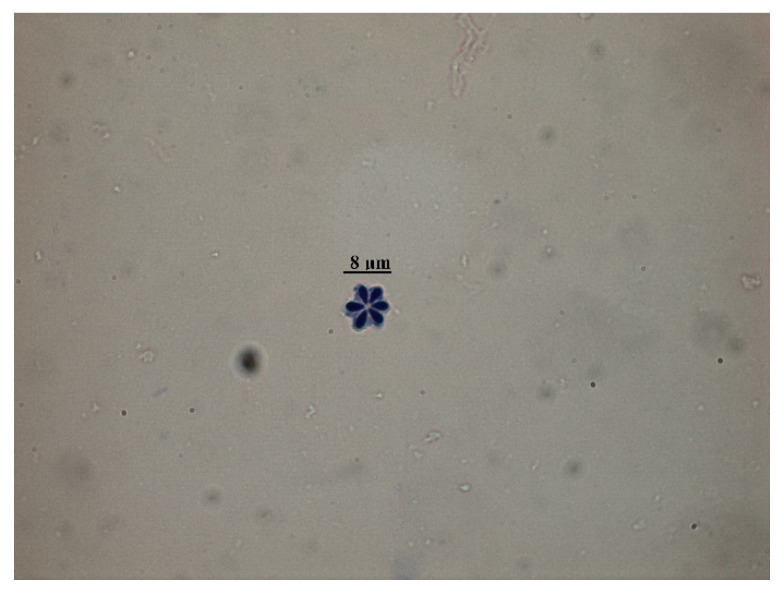
Morphological observation of *Kudoa hexapunctata* by muscle biopsy of Pacific bluefin tuna (*Thunnus orientalis*) cultured in South Korea. Scale bar = 8 μm.

**Figure 2 animals-10-01705-f002:**
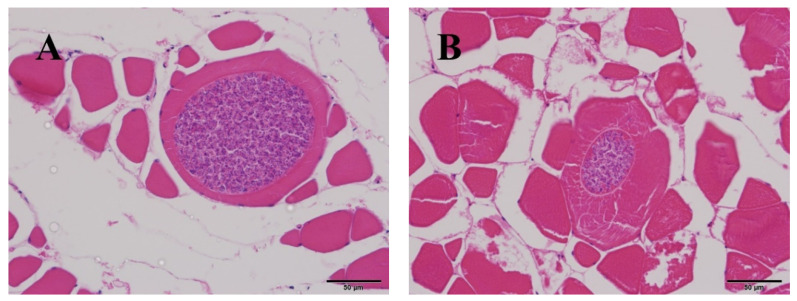
Histological analysis of abdominal and dorsal muscle in Pacific bluefin tuna (*Thunnus orientalis*) positively infected with *Kudoa hexapunctata*. H&E staining revealed: (**A**) abdominal muscle of the fifth sample; (**B**) abdominal muscle of the fifth sample. Scale bar = 50 μm.

**Figure 3 animals-10-01705-f003:**
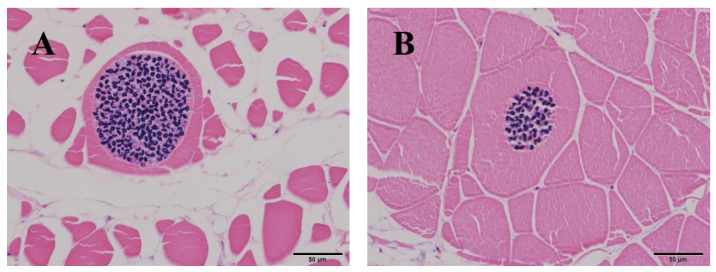
Histological analysis of abdominal and dorsal muscle in Pacific bluefin tuna (*Thunnus orientalis*) positively infected with *Kudoa hexapunctata*. Giemsa staining revealed: (**A**) abdominal muscle of the fifth sample; (**B**) abdominal muscle of the fifth sample. Scale bar = 50 μm.

**Figure 4 animals-10-01705-f004:**
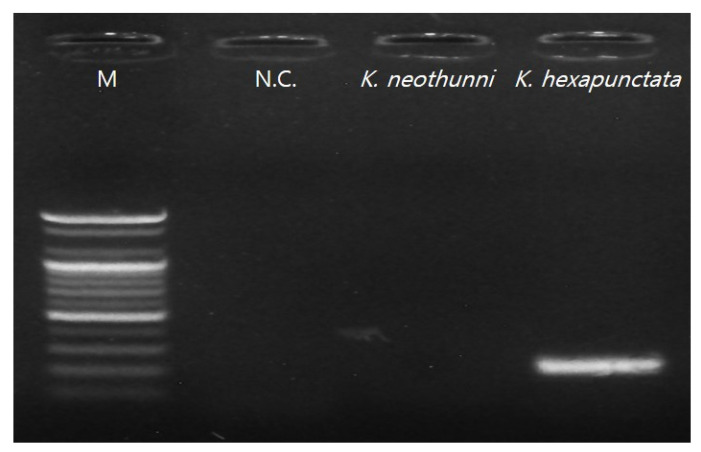
Electrophoresis results of secondary polymerase chain reaction (PCR) for distinguishing *Kudoa neothunni* from *Kudoa hexapunctata* (M = 2000 bp marker; N.C. = negative control).

**Table 1 animals-10-01705-t001:** Standards of condition factor (CF) used in trout and salmonid fish (Barnham and Baxter, [15]).

Condition Factor (CF)	Comments
1.60	Excellent condition, trophy class fish.
1.40	A good, well-proportioned fish
1.20	A fair fish, acceptable to many anglers.
1.00	A poor fish, long and thin.
0.80	Extremely poor fish, resembling a barracuda, with a big head and narrow, thin body.

**Table 2 animals-10-01705-t002:** Base sequence information and polymerase chain reaction (PCR) conditions of the primer set used in this study.

Primer	Sequences (5′–3′)	PCR Conditions	Length of Seq.	Reference
ERIB1	ACCTGGTTGATCCTGCCAG	30 cycles of 95 °C for 1 min, 48 °C for 1 min and 72 °C for 2 min, followed by 10 min incubation at 72 °C	-	Barta et al. [17]
ERIB10	CTTCCGCAGGTTCACCTACGG			
MyxospecF	TTCTGCCCTATCAACTWGTTG	30 cycles of 95 °C for 1 min, 52 °C for 1 min and 72 °C for 2 min, followed by 10 min incubation at 72 °C	769	Ivan Fiala [18]
MyxospecR	GGTTTCNCDGRGGGMCCAAC			
K. hexF	GCGACGGTTCAATCGTCAG	Incubation for 2 min at 94 °C, 35 cycles of 30 s at 94 °C, 1 min at 68 °C, and a final extension for 5 min at 68 °C	197	Arai et al. [6]
K. hexR	CACTGGTTGTTGAACTACCACACG			
K. neoF	GGCTACGTGCGGAAAAGTTGTA		1045	
K. neoR	CGCGGATTCCAACCTATATCCAG			

**Table 3 animals-10-01705-t003:** Biomass, condition factor (CF) and haematological data of currently analysed Pacific bluefin tuna (*Thunnus orientalis*) 1 to 5.

Parameters	1	2	3	4	5
Weight (kg)	17.4	17.8	14.5	10.4	13
Length (cm)	103	101	98	90	97
Condition factor (CF)	1.59	1.73	1.54	1.43	1.42
ALP (U/L)	170	190	382	185	367
BUN (mg/dL)	2.2	2.1	3	1.8	2
Ca (mg/dL) ^1^	9.5	10.9	12.8	9.4	10.2
GLU (mg/dL)	86	96	97	102	102
GOT (U/L)	453	347	563	295	354
GPT (U/L)	42	32	401	65	47
Haemoglobin (g/dL)	16.25	14.99	14.37	14.47	10.90
Haematocrit (%)	41	45	52	40	34
LDH (U/L) ^2^	503	393	497	314	287
TCHO (mg/dL)	141	141	204	204	197
TP (g/dL)	6.7	7.7	8.5	5.4	5.3

^1^ 2-fold diluted in PBS for analysis; ^2^ 10-fold diluted in PBS for analysis.

**Table 4 animals-10-01705-t004:** Sequence data of first and secondary polymerase chain reaction (PCR) amplicon.

*Kudoa hexapunctata* from Cultured Pacific Bluefin Tuna (*Thunnus orientalis*) in South Korea
(1) 769 bp (Ivan Fiala [18]; Barta et al. [17])
5′-TTCTGCCCTATCAACTAGTTGGTGAGGTAGTGGCTCACCAAGGTTGTGACGGGTAACGGGGGATCAGGGTTCGATTCCGGAGAGGGAGCCTGAGAAACGGCTACCACATCTAAGGAAGGCAGCAGGCGCGCAAATTACCCAATCCAGACTTTGGGAGGTAGTGACGAGAAATACCGGAGTAGACCGTTAATTGGTTCACTATCGGAATGAACGTAATTTAATACCTTCGATGAGTAGCTACTGGAGGGCAAGTCTGGTGCCAGCAGCCGCGGTAATTCCAGCTCCAGTAGTGTATATCAAAATTGTTGCGGTTAAAACGCTCGTAGTTGGATTACAAAAGCTCTTTGGCGGTTAAATCAAGGTTTGATCGCTGTGGGGTTTTTTTATCGCGAGAGCCGCACGTGGGATTAAATTCTTGTGTGTGGTCACTTGCGAGGTGTGCCTTGAATAAAGCACAGTGCCCAAAGCAGGCGTAAGCTTGAATGTTATAGCATGGAACGATTATGTTAATCTTGTCGACTGTTGGTTGTTGGCAGTGGTCTCGATTAAAAGGGACATTTGAGGGCGTTAGTACTTGGTGGCGAGGGGTGAAATCCTTAGACCCATCAAAGACTAACTAATGCGAAAGCATTCGCCAAGAGTGTTTTCATTAATCAAGAACGAAAGTTGGAGGTTCGAAGACGATCAGATACCGTCCTAGTTCCATACAGTAAACTATGCCAACATGGGATTAGCCCGGTTTAATCCAGGTTGGTCCCTCAGAGAAACC-3′
(2) 197 bp (Arai et al. [6])
5′-CACTGGTTGTTGAACTACCACACGACTTTTCCGCACGTAGCCACAATGTGACCAGCACGAAACGCCGCACAGCACCAACCGACCATACATCACCAACGCTCCATCACATACATATCAGCACACGAGGCGGATCACCAGCCGCAACCACGAAACATTTGAGTGTTCCGTGACAGCCATGCTGACGATTGAACCGTCGC-3′

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
