# Peer review of "The First Detection of Kudoa hexapunctata in Farmed Pacific Bluefin Tuna in South Korea, Thunnus orientalis (Temminck and Schlegel, 1844)"

_animals, 2020, doi:10.3390/ani10091705_

Round 1
Reviewer 1 Report
I personally suggest to rewrite the manuscript in form of short communication reporting only the first finding of Kudoa hexapunctata in Thunnus orientalis, without speculating on possible consumer risk.
The authors should refer to the annotated PDF for specific comments.

Author Response
Dear reviewer 1,
"Please see the attachment."
Thank you very much for taking a close look at my manuscript. We tried our best to reflect all the opinions of the reviewer 1, but unfortunately, there are a few parts that were not reflected.
Particularly, the part about gross pathology lesions have helped me a lot. I would like to say thank you very much.
We have attached a section on the revised matters, and we hope that this manuscript will satisfy the reviewer's comments.
My best regards,
Gyoungsik Kang

Reviewer 2 Report
Dear Authors,
I congratulate you for your manuscript, that is really well written, with good style and fluency, and an average interest for the reader. The topic treated is important for more reasons related not only to the studied area, but, considering the distribution of Kudoa hexapunctata I consider essential but right the choice of deal only your geographical area and related species.
However, if you can find other related good references to complement introduction section, it could be richer.
About the weakness that i found, there are few minor points:
- A few number of sample used, that don't give in my opinion the right soundness that this manuscript could have.
- Fig 6 needs a legend.
- From haematological parameter detected, there is in the positive sample a low level of LDH, this maybe deserves to be discussed, despite what you say yourself the result was no statistically significant cause the little number of sample.
There is in my opinion only a major point to assess in this manuscript, related to the title and consequently to extent to the entire document. You have entitled and present this manuscript as "The first detection of Kudoa hexapunctata in cultured Pacific bluefin tuna, Thunnus orientalis", without a relationship with studied area. But checking the references, I found in [10] the real "first detection" of Kudoa hexapuctata on some farmed specimens of Tunnus orientalis (from Japanese area), so your manuscript absolutely cannot be considerated such as the overall first detection, but it must be related to the sampling area, because, as you can see on Yokoama et al., Kudoa hexapunctata was already previously reported on farmed Tunnus orientalis specimens. I think that, before pubblication, this point must be fixed in the title and in the entire manuscript (as already done on Conclusions section), to frame it correctly for the readers and scientific community.
Best regards
Author Response
Dear reviewer 2,
"Please see the attachment."
Thank you very much for taking a close look at my manuscript. We tried our best to reflect all the opinions of the reviewer 1, but unfortunately, there are a few parts that were not reflected.
We have attached a section on the revised matters, and we hope that this manuscript will satisfy the reviewer's comments.
My best regards,
Gyoungsik Kang

Round 2
Reviewer 1 Report
This new version has been improved and resubmitted as brief report. Therefore many of my previous concerns related to a regular research paper have been overcome in this amended and simplified version.
Nevertheless, I suggest the authors to stress in the "introduction" and "conclusions" sections the preliminary value of the manuscript, needing further research and confirmation with an adequate statistical design.
Moreover, I suggest the authors to seek advice from a native English speaker.
Author Response
Dear Reviewer 1,
Thank you for your letter,
We have done our best to reflect your comments.
Point 1. I suggest the authors to stress in the "introduction" and "conclusions" sections the preliminary value of the manuscript, needing further research and confirmation with an adequate statistical design.
Response 1. We have added sentences at the end of the "Introduction" and "Conclusions". "Furthermore, through this study, we intend to provide a resource to suggest K. hexapunctata infection in South Korean farmed tuna and to analyse the correlation between the source of infection and food poisoning." (Introduction 64-66)
"Therefore, the further study needs to test for potential pathogenicity in laboratory mammals for the purpose of speculating about possible foodborne disease." (Conclusions 192-194)
Point 2. I suggest the authors to seek advice from a native English speaker.
Response 2. This manuscript already edited by Elsevier Language Editing Services. We attached a PDF file to prove this.
My best regards

Reviewer 2 Report
Dear Authors,
after the changes made after the first review, I think that now the manuscript is ready to be accepted in the present form.
Best regards
Author Response
Dear Reviewer 2,
Thank you for your letter,
We can learn a lot from your reviews.
I would like to express my sincere thanks.
And also, as an additional note, we have edited this manuscript through Elsevier Language Editing Services. We attached a PDF file to prove this.
My best regards
